# Optimizing and Predicting Antidepressant Efficacy in Patients with Major Depressive Disorder Using Multi-Omics Analysis and the Opade AI Prediction Tools

**DOI:** 10.3390/brainsci14070658

**Published:** 2024-06-28

**Authors:** Giulio Corrivetti, Francesco Monaco, Annarita Vignapiano, Alessandra Marenna, Kaia Palm, Salvador Fernández-Arroyo, Eva Frigola-Capell, Volker Leen, Oihane Ibarrola, Burak Amil, Mattia Marco Caruson, Lorenzo Chiariotti, Maria Alejandra Palacios-Ariza, Pieter J. Hoekstra, Hsin-Yin Chiang, Alexandru Floareș, Andrea Fagiolini, Alessio Fasano

**Affiliations:** 1Department of Mental Health, Azienda Sanitaria Locale Salerno, 84123 Salerno, Italy; g.corrivetti@aslsalerno.it (G.C.);; 2European Biomedical Research Institute of Salerno (EBRIS), 84125 Salerno, Italy; 3Protobios, 12618 Tallinn, Estonia; 4Centre for Omic Sciences, Joint Unit Eurecat Technological Centre of Catalonia-Rovira i Virgili University, Unique Scientific and Technical Infrastructure (ICTS), 43204 Reus, Spain; salvador.fernandez@eurecat.org; 5Mental Health Research Group, Institut d’Investigació Biomèdica de Girona-CERCA, 17190 Girona, Spain; eva.frigola.ias@gencat.cat; 6Mental Health and Addictions Network, Institut Assistència Sanitària (IAS), 17190 Girona, Spain; 7Perseus Biomics BV, 3300 Tienen, Belgium; 8Biokeralty Research Institute AIE, 01510 Vitoria-Gasteiz, Spain; 9Department of Psychiatry, Faculty of Medicine, Istanbul Medipol University, 34214 Istanbul, Turkey; 10Mama Health Technologies GmbH, 14482 Potsdam, Germany; 11Ceinge Biotecnologie Avanzate SCRL, 80131 Napoli, Italy; 12Unidad de Investigación, Fundación Universitaria Sanitas, Bogotá 110811, Colombia; 13Department of Child and Adolescent Psychiatry, University Medical Center Groningen, University of Groningen, Hanzeplein 1, 9713 GZ Groningen, The Netherlands; p.hoekstra@accare.nl; 14Accare Child Study Center, 9723 HE Groningen, The Netherlands; 15Cephalgo SAS, 67000 Strasbourg, France; 16Artificial Intelligence Expert (AIE), 400310 Cluj-Napoca, Romania; 17Department of Molecular and Developmental Medicine, Division of Psychiatry, University of Siena School of Medicine, 53100 Siena, Italy; andrea.fagiolini@unisi.it; 18Department of Pediatric Gastroenterology and Nutrition, Mucosal Immunology and Biology Research Center, Massachusetts General Hospital, Boston, MA 02114, USA; 19Department of Pediatrics, Harvard Medical School, Harvard University, Boston, MA 02138, USA; 20Mucosal Immunology and Biology Research Center, Massachusetts General Hospital, Harvard Medical School, Boston, MA 02114, USA

**Keywords:** major depressive disorders, microbiome, metabolomic, transcriptomics, inflammation, genetics, artificial intelligence, EEG, chatbot, personalized medicine

## Abstract

According to the World Health Organization (WHO), major depressive disorder (MDD) is the fourth leading cause of disability worldwide and the second most common disease after cardiovascular events. Approximately 280 million people live with MDD, with incidence varying by age and gender (female to male ratio of approximately 2:1). Although a variety of antidepressants are available for the different forms of MDD, there is still a high degree of individual variability in response and tolerability. Given the complexity and clinical heterogeneity of these disorders, a shift from “canonical treatment” to personalized medicine with improved patient stratification is needed. OPADE is a non-profit study that researches biomarkers in MDD to tailor personalized drug treatments, integrating genetics, epigenetics, microbiome, immune response, and clinical data for analysis. A total of 350 patients between 14 and 50 years will be recruited in 6 Countries (Italy, Colombia, Spain, The Netherlands, Turkey) for 24 months. Real-time electroencephalogram (EEG) and patient cognitive assessment will be correlated with biological sample analysis. A patient empowerment tool will be deployed to ensure patient commitment and to translate patient stories into data. The resulting data will be used to train the artificial intelligence/machine learning (AI/ML) predictive tool.

## 1. Introduction

Nearly one billion people worldwide suffer from a mental illness, with MDD being the second most frequent mental disorder worldwide, after anxiety disorder [1]. MDD is characterized by decreased energy and interest in daily activities, feelings of sadness, difficulty concentrating, sleep disturbances, feelings of emptiness, irritability, or suicidal thoughts, plans, and attempts. Over 700,000 suicides are recorded each year, making major depression the fourth leading cause of death in people aged 15–29 years [2]. Onset occurs during adolescence in nearly 50% of cases, and there is a high risk of recurrence and chronicity across the lifespan in 5% of cases. The percentage of adults who have experienced some symptoms of depression is highest in the population aged 15–29 years (21%), followed by those aged 45–64 years (18.4%) and 65 years and over (18.6%), and lastly those aged 30–44 years (16.8%) [3]. Global lifetime prevalence is about 10 to 15%, and some studies estimate this prevalence to be between 8.1 and 11.2% in low/medium-income countries and 13% in high-income countries [4,5]. Studies that have explored the prevalence and distribution of MDD in African Americans, non-Hispanic Caucasian Americans, and Caribbean African Americans have found that the overall lifetime prevalence of MDD disorder among Caucasians is 17.9% as opposed to African Americans, whose prevalence estimate is only 10.4%. The difference between African Americans and Caucasians lies in the fact that the chronicity of disease is higher for African Americans (56%) than it is for Caucasian patients (38.6%) [6]. The countries with the highest rates of MDD are located in central Europe, eastern Europe, central Asia, Latin America, and the Caribbean [7].

MDD is a multifactorial disease caused by a combined effect of genetic, epigenetic, psychological, biological, and environmental factors. It is also driven by compositional and functional changes in the gut microbiome [8]. The microbiome exerts its effect on the nervous system, the so-called microbiome–gut–brain (MGB) axis, through the synthesis of metabolites (e.g., short-chain fatty acids (SCFAs) and key dietary amino acids (such as tryptophan (TRP)) that induce neurotransmitter production (e.g., serotonin, gamma-aminobutyric acid (GABA), and glutamate) and immune system activation [9,10,11]. In addition to influencing specific brain functions, there is now growing evidence that the gut microbiome may also affect epigenetic patterns and indirectly influence the efficacy of several drugs, including antidepressants. In fact, specific DNA methylation signatures have recently been proposed for assessing MDD risk and predicting drug response [12,13]. The criteria for the diagnosis of MDD are included in the Diagnostic and Statistical Manual of Mental Disorders Fifth Edition Text Revision (DSM-5-TR) and the International Classification of Diseases, 11th Revision (ICD-11). Both diagnosis and treatment of MDD remain challenging due to the lack of early biomarkers and personalized treatment options, especially for pediatric patients who are still considered “little adults”. Recommended and commonly used treatments for depression typically include antidepressants, such as selective serotonin reuptake inhibitors (SSRIs), and/or psychotherapy, including cognitive behavioral therapy (CBT). On average, it takes at least four weeks to see a response to medication and six weeks to reach remission. However, remission can last longer than 12 weeks, and remission is rarely achieved with the first drug prescribed [14]. Even the most advanced treatments often fail to produce long-lasting results, with about 40–50% of patients experiencing a relapse within 1–2 years of treatment. Remarkably, about one-third of people with MDD fail to achieve complete remission of symptoms even after two adequate antidepressant trials [15]. To address these issues, several newly developed and emerging interventions are being evaluated in clinical trials or have been introduced into clinical practice for the treatment of severe and/or treatment-resistant MDD [16]. An additional barrier to appropriate care of depressed patients is the lack of tools to monitor symptoms and assess adherence and efficacy of treatments, particularly in identified cases of treatment-resistant depression routinely and remotely.

### Objective

OPADE is a non-profit, observational, multicenter, open-label study aimed at identifying predictive biomarkers for stratification and implementation of personalized drug treatments in patients with MDD, to guide healthcare provider decision-making by developing an AI/ML predictive tool that will be capable of combining genetics, epigenetics, microbiome, immune response data, and also the non-molecular biomarkers such as medical history, electroencephalography (EEG), from subjects with MDD.

## 2. Study Design

The project is led by the European Biomedical Research Institute of Salerno (EBRIS), Salerno, Italy, in collaboration with five clinical centers located in different countries (UNISI—Siena Italy, FUS—Bogotá Colombia, ACC—Groningen The Netherlands, IDIBGI—Groningen, Spain, MED—Instambul, Turkey). The study protocol includes six follow-up visits: T0 (enrollment), T1, T2, T3, T4, and T5. At each medical visit, psychometric scales will be administered to the patients, and contextually biological samples, including blood, stool, and saliva, will be collected, as shown in Figure 1. The study will use a multi-omics approach, including metagenomic sequencing to characterize the microbiome composition, metabolomics to detect circulating metabolites, transcriptomics to quantify microRNAs, and epigenomics to assess methylation variability between and within groups and immune assays to analyze the antibody immune response and inflammatory profiles (cytokines, interleukins, and growth factors). Cortisol and lipoproteins will also be quantified. In parallel, cognitive assessment and emotional status will be recorded remotely by each patient via chatbot and wearable EEG devices, respectively. With the chatbot, the enrolled can share her/his feelings or significant patient journey events happening during the trial with the other participants, and the chat conversations will be translated into standardized data. The EEG device also allows patients to document their instant emotions via the accompanying mobile app and record their corresponding brainwaves.

### 2.1. Setting and Participants

Three hundred and fifty patients diagnosed with MDD will be enrolled for 24 months and divided into 4 groups according to age: 14–17 years (70 pediatric patients), 18–30 years (100 adult patients), 31–39 years (90 adult patients), 40–50 years (90 adult patients). All data will be collected in a dedicated and protected electronic database called REDCap 14.3.14, based on the use of standardized electronic case report forms (eCRFs). Patients who meet all inclusion criteria and none of the exclusion criteria are eligible for the study. Inclusion and exclusion criteria are listed in Table 1.

### 2.2. Serological Markers

Multiplexed immunoassays will be performed to analyzed and evaluate several inflammatory markers including G-CSF, GM-CSF, IFN-γ, IL-10, IL-12p40, IL-15, IL-1α, IL-1β, IL-2, IL-4, IL-5, IL-6, IL-8/CXCL8, MCP-1/CCL2, TNF-α, and TNFβ/lymphotoxin-a. Cytokine concentration is measured by a Bio-Plex 200 system multiplexing instrument (Bio-Rad, Hercules, CA, USA) using the MILLIPLEX MAP Human Cytokine/Chemokine Magnetic Bead Panels (Merck Millipore, Darmstadt, Germany). Signals will be detected using the Bio-Plex Manager 6.2 software. Mimotope variation analysis (MVA), a new generation random peptide phage display method, will be used to delineate antibody epitope profiles from blood plasm with high precision. In brief, plasma samples will be subjected to MVA [17,18]. For that, a modified M13 phage library displaying 10 × 109 different random 12-mer peptide sequences on pIII coat protein will be incubated with plasma immunoglobulin Gs (IgGs). The phage-IgG complexes will be captured to protein G-coated magnetic beads, the unbound phage particles will be eliminated by frequent washes, and the bound fraction will be subjected to DNA purification. For DNA sequencing, unique barcodes will be added to each sample using specific primers and the PCR amplification method. Amplified PCR products will be quantified, normalized, pooled, and sequenced with Illumina NextSeq2000 systems (Illumina Italy S.r.l. Viale Certosa 218 Quartiere Garegnano, 20156 Milano, Italy). Obtained sequences will be translated into 12 mer peptides and shared epitopes in peptides will be delineated by further data analyses.

### 2.3. Lipoprotein Profiling

For lipoprotein profiling, we will use the Car–Purcell–Meiboom–Gill (CPMG) pulse sequences on a 600 MHz Bruker Avance III NMR spectrometer (Bruker BioSpin, Zurich, Switzerland). Different forms of lipoproteins will be evaluated: apolipoproteins A1 and A2, HDL-apolipoprotein A1 and A2, HDL3 free cholesterol, HDL3-apolipoprotein A1, HDL2-apolipoprotein A2, apolipoprotein A2, IDL, HDL-apolipoprotein A2, VLDL and its subtypes, VLDL2- triglycerides, VLDL3-triglycerides, VLDL2-cholesterol, VLDL3-cholesterol, VLDL4-cholesterol, VLDL4-free cholesterol, VLDL2-phospholipids, VLDL3-phospholipids, LDL5-cholesterol, LDL5-free cholesterol, LDL5-phospholipids, LDL5-apolipoprotein B, HDL3-cholesterol, HDL4-cholesterol, free cholesterol HDL3, HDL4 free cholesterol, HDL3-phospholipids, HDL4-phospholipids, HDL3-apolipoprotein A1, HDL4-apolipoprotein A1, HDL3-apolipoprotein A2, and HDL4-apolipoprotein A2.

### 2.4. Microbiome Profiling

DNA will be extracted from stool samples with the QIAGEN PowerFecal Pro DNA extraction kit (QIAGEN, Venlo, The Netherlands). DNA concentration will be quantified using a Nanodrop instrument. Libraries will be prepared using an Illumina DNA prep kit (Illumina, San Francisco, CA, USA). Extracted DNA will be indexed with Illumina DNA/RNA UD Indexes Set A, Tagmentation, and samples pooled. In total, 150 bp pair-ended sequencing will be performed on an Illumina NextSeq 1000 platform. Negative controls will be prepared during DNA extraction and will be sequenced to ensure the identification of potential bacterial contaminations from working environments and the kits in use. We aim to perform sequencing at a depth of 20 million reads per sample. The resulting sequencing reads will be quality-filtered with TRIMMOMATIC. Reads will be assigned to bacteria, fungi, and viruses using MetaPhlan v4 [19]. The abundance of bacterial metabolic pathways will be identified with HUMAnN3 v3.8 [20].

### 2.5. Transcriptomics

Total RNA will be extracted using a QIAGEN RNeasy Mini Kit following the manufacturer’s instructions, and microRNA (miRNA) expression assay will be performed with NanoString nCounter (NanoString Technologies; Seattle, WA, USA). The assay is designed to provide an ultrasensitive, reproducible, and highly multiplexed method to detect miRNAs in total RNA at all biological levels of expression without the use of reverse transcription or amplification.

### 2.6. Epigenomics

We will perform methylome analysis of genomic DNA extracted by blood and saliva with QIAGEN, Puregene Blood Core Kit A. The Epic Array Illumina platform will be used to interrogate the methylation state of 850,000 CpG sites per sample. Depression-related genes and CpG sites will be bioinformatically extracted and analyzed.

### 2.7. Metabolomic Profile

Metabolomic analyses will be performed on stool samples with a GC-MS system (GC-2010 Plus gas chromatograph and QP2010 Plus mass spectrometer; Shimadzu Corp., Kyoto, Japan). The identification of metabolites will be conducted by using external standards according to the Level 1 Metabolomics Standards Initiative (MSI) [21].

### 2.8. Pharmacogenetic and Long qt Phenotype

Genetic analysis will be conducted through a saliva sample. Cardiovascular risk factors will be obtained through a blood sample, electrocardiogram, and data extracted from clinical records (BMI, weight, and height). Socio-demographics, clinical and social functioning outcomes (TSES, UFS, Cognitive scales) QoL, and lifestyle will be self-reported by patients.

### 2.9. Hormonal/Cortisol Analysis

A saliva sample will be collected from the patient to quantify cortisol using the enzyme-linked immunosorbent assay (ELISA) technique. Saliva samples will be incubated with a tracer for 30 min at 37 °C. Detection will be performed using a gamma counter. The lower detection limit of the assay is 0.8 nmol/L.

### 2.10. Psychometric Rating Scales

In optimizing the administration of rating scales in our study, several strategies will enhance participant experience and data quality. First, implementing a rotation of scales across participants will help to distribute cognitive load and prevent monotony-induced fatigue. Additionally, incorporating breaks between scale administrations allows participants to recharge mentally, mitigating cognitive overload. Moreover, soliciting feedback from participants about their experience with the scales, including fatigue levels and suggestions for improvement, fosters participant engagement and improves protocol effectiveness. Creating a supportive environment with comfortable seating, adequate lighting, and privacy minimizes distractions and promotes focus during scale administration. Lastly, providing comprehensive training and ongoing support to research staff ensures consistent and accurate scale administration, reducing the need for repeated assessments and enhancing data reliability. By implementing these strategies, we aim to optimize the rating scale administration process, ultimately enhancing the validity and reliability of our study outcomes. The Hamilton Rating Scale for Depression (HAM-D) is a questionnaire designed for adults to assess and measure the severity of depression [22]. It consists of multiple items that assess various aspects, including mood, guilt, suicidal ideation, sleep patterns, agitation or retardation, anxiety, weight changes, and somatic symptoms. The HAM-D serves as a tool to indicate the presence of depression and is also used to guide recovery assessments. The Beck Depression Inventory-Second Edition (BDI-II) is a widely used self-report inventory that assesses the severity of depression in adolescents and adults over a two-week period. It has been shown to be valid across diverse populations and cultures [23].

The Montgomery–Åsberg Depression Rating Scale (MADRS) is a diagnostic questionnaire used to measure the severity of depressive episodes in patients with mood disorders [24]. It is more sensitive than the Hamilton Scale to changes induced by antidepressants and other forms of treatment. The Mood Spectrum Self-Report-Current (Mood_SR_C, Mood_SR last month) is a psychometric questionnaire that assesses a broad range of mood psychopathology traits [25]. These include core DSM symptoms of depression and mania, subthreshold manifestations, mood-related personality traits, prodromal and residual symptoms, and behaviors associated with mood disorders or developed as coping mechanisms.

### 2.11. Assessment of Personal Resources

The Toronto Side Effect Scale (TSES) is a clinical interview side-effect questionnaire that assesses both the frequency and the severity of common adverse side effects. The Internalized Stigma of Mental Illness (ISMI) assesses the internalized experience of stigma and self-evaluation [26]. The Service Engagement Scale (SES) explores the relationship with mental health services.

### 2.12. Assessment of Real-Life Functioning and Quality of Life

The Global Assessment of Functioning (GAF) is a widely used measure of illness severity that considers psychological, social, and occupational functioning [27]. It provides a single score or separate scores for symptoms (GAF-S) and functioning (GAF-F).

The Childhood Global Assessment Scale (CGAS) [28] measures global functioning, while the Global Functioning: Social Scale (GF-Social) and the Global Functioning: Role Scale (GF-R) independently assess social and role functioning.

The Short Form 36 for Adults (SF36) is a 36-item questionnaire that measures several health-related qualities of life variables [29].

The Pediatric Quality of Life Inventory (PedsQL) is a general health status instrument that assesses health domains in children and adolescents, including physical, emotional, psychosocial, social, and school functioning [30].

The International Physical Activity Questionnaire (IPAQ) is a short 7-item questionnaire for the assessment of physical activity. It has been validated and used in different population settings for monitoring physical activity, as reported by the patient, between 18 and 65 years of age.

KIDSCREEN 27 is a short questionnaire completed by the adolescent patient to assess quality of life in childhood across 5 domains (physical activity, mood, family, friends, and school) [31]. The 14-item questionnaire assesses adherence to the Mediterranean diet by assigning a score to each of the items. The International Physical Activity Questionnaire (IPAQ) will be used to register patterns of physical activity and the PREDIMED (PREvention with MEDiterranean Diet) scale will register adherence to Mediterranean dietary patterns [32].

The Social Functioning Scale consists of seven subscales: (1) social engagement/withdrawal (amount of time to spend alone, the likelihood to initiate conversation); (2) interpersonal behavior (number of friends, engagement in a romantic relationship); (3) prosocial activities (participation in social activities, e.g., visit friends, play sports); (4) recreation (engagement in activities and hobbies); (5) independence–competence (ability to maintain independent living); (6)independence–performance (performance of the skills required for independent living); (7) employment/occupation (engagement in employment), although the total score is typically used to provide an overall social functioning level.

### 2.13. Treatment-Resistant Depression

Treatment-resistant depression (TRD) refers to a lack of response to adequate trials of medications or other antidepressant treatments. Studies suggest that 30–50% of patients with major depressive disorder (MDD) do not respond to initial antidepressant trials, and approximately 20% remain depressed up to 2 years after onset [33]. The Maudsley staging method (MSM) is a recent staging approach for TRD that views treatment resistance as a continuum influenced by multiple factors. It is a multidimensional method that considers clinical and treatment factors.

### 2.14. Assessment of Cognitive Functions

Cognitive deficits are common in people with depression, affecting both “cold cognition” (neutral information processing, planning, and cognitive flexibility) and “warm cognition” (misinterpretation of social interactions, prejudice, and negative attributional styles). Neuropsychological assessment tools reveal significant deficits that contribute to worsening depressive symptoms, impaired psychosocial functioning, and reduced quality of life. Some cognitive deficits persist beyond the acute phase, posing a risk of relapse, while others, particularly those related to “warm cognition”, may disappear during remission. Certain cognitive changes may serve as markers of the disease, possibly detectable before the first depressive episode, and may be associated with a long-term risk of developing dementia. It is important to investigate the relationship between cognitive performance and other biological markers of depression. Impaired domains include information processing speed, working memory, verbal memory, verbal fluency, executive function, and emotion recognition. The instrument used to assess cognitive performance will be the Cambridge Neuropsychological Test Automated Battery (CANTAB). The CANTAB is a battery of computerized neuropsychological tests that measure several cognitive domains [34,35,36]: sensitivity to feedback, emotion recognition, executive function, sustained attention, and episodic memory. The tests in this battery have been shown to discriminate between the neuropsychological profiles of bipolar disorder and unipolar depression. They are also sensitive to the effects of various interventions on patients. Cognitive performance on these tests correlates with quality of life, making the battery not only biologically but also clinically relevant to mood disorders. The CANTAB test battery was translated into Italian, Spanish, South American Spanish, Turkish and German.

### 2.15. Electroencephalographic Evaluation

The project includes an electroencephalographic evaluation using a non-surgical and non-invasive EEG tracker provided by Cephalgo. This device monitors and records brain electrical activity to identify potential anomalies associated with mental health conditions. The Mood Tracker for OPADE, utilizing EEG data, is not classified as a Medical Device but interprets emotional states for non-medical purposes like personal well-being. The distinction is essential as it aligns with the trend of non-MDR devices in medical research. The Mood Tracker for OPADE contributes supplementary data for research but does not provide medical information or directly impact medical decision-making processes.

### 2.16. Evaluation of Antidepressant Response

Metabolic pathways involved in patients’ response to antidepressants will be investigated in blood by liquid chromatography (UHPLC) coupled to triple quadrupole mass spectrometry (QqQ-MS) using a 1290 Infinity II UHPLC coupled to a 6490 QqQ-MS (Agilent Technologies, Palo Alto, CA, USA). The metabolomic analysis will include three distinct groups of metabolites: (i) tryptophan intermediates, including tryptophan, serotonin, 5-HIAA, kynurenine, kynurenic acid, and other hormones and derivatives involved in the pathway, and other related purines such as paraxanthine/xanthine ratio; (ii) L-acylcarnitine including short, medium and long chain acylcarnitines with a particular focus on laurylcarnitine and acetylcarnitine; (iii) phenolic and related compounds, including phenolic acid, mandelic acid or methoxy-hydroxyphenyl-glycol.

### 2.17. Remission Assessment

Within the 4 patient groups, OPADE will compare remission and non-remission of symptoms at each timepoint: T1, T2, T3, T4, and T5. The commonly used criteria for remission in clinical trials are based on threshold scores, or cut-offs, of standardized scales. Complete remission is defined as a relatively short period during which the individual is asymptomatic. Being asymptomatic is not defined as the complete absence of symptoms but as the absence of minimal symptoms. According to the literature, remission is defined as a score ≤ 7 on the 17-item Hamilton Depression Rating Scale (HAM-D) [37]. Among the group that did not achieve remission, the percentage of patients who received antidepressant potentiation with another antidepressant drug of a different class or with other drugs was also assessed.

## 3. Device Used in the Study

### Digital Patient Empowerment Tool: Turning Stories into Data

The study involves the use of a chatbot (accessible from any device), through which enrolled patients will be able to interface on a regular basis. For the duration of the study, they will be invited to participate in a virtual lobby moderated by professionals to share their stories and support each other. Indeed, peer support groups have been shown to reduce states of anxiety, helplessness, confusion, and depression [38]. Through mathematical models, patients’ stories will be transformed into data accessible to the entire research group [39,40,41]. Traditionally, patient narratives have been invaluable qualitative data in healthcare research, yet analyzing and sharing this information effectively across teams has posed challenges. However, a groundbreaking approach has emerged: harnessing mathematical models to bridge this gap.

By converting patients’ stories into quantitative data, a new level of comprehension is unlocked. These models capture qualitative elements such as emotions and challenges as measurable data points, enabling collaborative analysis among diverse research team members.

Picture psychologists, data scientists, and clinicians will work together to analyze quantitative data derived from patient narratives. This not only enhances data accessibility but also fosters communication and collaboration within the research group.

## 4. Factors of Interest

### 4.1. Environmental

The ensemble of environmental factors, such as childhood adversity, stressful life events, and socioeconomic status, may have cumulative effects on major depressive disorder. Harmful determinants can also be derived from dietary habits. Indeed, dietary components (e.g., vitamin, fiber, folate, zinc, selenium, and iron intake) and diet-related inflammatory potential have been linked to microbiome composition and, ultimately, to depression outcomes. Although numerous studies have attempted to examine the interplay of psychological aspects with environmental risk factors and biological mechanisms, the pathways that contribute to the onset of MDD remain far from elucidated [42]. This lack of progress is partly due to the complexity and clinical heterogeneity of depression, combined with the analytical inconsistency of the literature, which does not allow the identification of the theoretically involved biomarkers with sufficient proven specificity, sensitivity, and reproducibility. Moreover, biomarker research in psychiatry has proven to be more complex than in other medical disciplines, and it is still debated whether mental disorders should be conceptualized as discrete entities (categorical approach) or as phenomena along a continuum of severity (dimensional approach).

### 4.2. Genetics and Epigenetics

Major depressive disorder cannot be attributed to a single factor but rather to the combination of many variables that contribute to the onset of the disease based on the underlying genetic characteristics of the patients. In many cases, environmental exposures have been shown to specifically contribute to disease, but not all exposed individuals experience depressive symptoms. Similarly, not all individuals who carry a specific genetic predisposition become patients. This simple evidence suggests the need for a synergistic interaction of external factors and genetic predisposition to increase disease risk. To date, epigenetic modification has been shown to be one of the strongest associations with disease. For example, specific methylation status in CpG islands of BDNF, HTR1A, and HTR1B, IL11, and IL6 genes have been associated with antidepressant response. Examples of the effects of the sample size and of how these compare to other biological measures have been recently and extensively reviewed [43].

However, most studies suffered from small sample size, or cell heterogeneity, or lacked cross-analysis with additional biomarkers.

### 4.3. Microbiome/Metabolome

The microbiota–intestine–brain (MIB) axis is involved in a bidirectional communication that plays a critical role in the normal development of the immune, endocrine, and nervous systems. Exposure to various types of stress has been reported to alter the composition of the microbiota, which in turn affects responses to stress and anxiety. Stress can lead to increased intestinal permeability and inflammation [44,45]. Compounds produced by gut bacteria or inflammatory processes may also affect nervous system function. Measures of different immune markers have been found to represent reliable biomarkers of disease and are useful for therapy decisions. In addition to influencing specific brain functions, there is now increasing evidence that the gut microbiota could also influence the efficacy of several antidepressants by enzymatically metabolizing drugs. Therefore, to implement personalized medicine and validate specific biomarkers for patients’ stratification and prediction of drug efficacy, establishing the composition and metabolic function of the gut microbiota is becoming a key asset.

## 5. Statistical Approach

The analysis will be conducted using descriptive statistics and statistical models. Summary statistics for continuous variables will include mean, standard deviation, median, and minimum/maximum range. Categorical variables will be presented as frequency counts and percentages. As OPADE is a prospective project, variables distributed over time will be summarized using Kaplan–Meier plots, highlighting the median and confidence intervals (CI). The comparison between two or more results of the analysis will be reported: The incidence rate, the confidence interval, and the difference between the confidence intervals (with CI ≥ 95%; *p* < 0.05). The CI will be calculated using the test-based method of Sahai H, Khurshid A (1996) [46], while the *p* value will be obtained using the chi-squared (χ2) statistical model. The Pearson correlation coefficient is used to infer relationships between biological and clinical indices. For the analysis of epigenomic data, we will perform bioinformatic analysis using RnBeads scripts in R software v4.4.1 [47]. DNA methylation distributions will be analyzed, and inter- and intra-group variability in methylation profiles will be quantified. In addition, differential methylation between groups of samples will be characterized [48]. Differentially methylated CpG sites, promoters, and CpG islands will be calculated between individual samples and between groups using Mann–Whitney tests. Principal component analysis (PCA) will be performed according to the dissimilarities in DNA methylation at each of the 850 k CpG sites, and PCA plots will be generated. Metagenomic data will be analyzed using the R package vegan. We will investigate the bacterial community composition at all taxonomic levels (from phylum to species). We will calculate species diversity and richness and compare bacterial composition between groups using permutational multivariate analysis of variance (PERMANOVA) tests. Differences in the relative abundance of species between groups of samples will be further investigated using PCA. Differentially abundant taxa will be identified using Mann–Whitney tests. Metabolomics data will be analyzed using appropriate statistical tests (paired or non-paired), and individuals will be stratified by different chemometric tools such as PCA, partial-least square discriminant analysis (PLS-DA), heatmap, and correlation analysis. The suitability of specific biomarkers will be analyzed using receiver operating characteristic (ROC) curves when necessary.

### 5.1. Artificial Intelligence Modelling

Text analysis with deep neural networks (DNNs) involves preprocessing textual data through tokenization, cleaning, and word embedding to convert it into numerical form [49]. The DNN architecture, with its multiple hidden layers and activation functions, is meticulously designed to extract intricate patterns and relationships within the data [50]. This architecture is tailored based on dataset characteristics and specific task requirements. DNNs excel in capturing complex non-linear relationships, making them suitable for text analysis tasks. They can produce probability distributions for classification tasks and numerical values for regression tasks. Additional techniques like batch normalization and hyperparameter tuning are employed to enhance performance and stability during training [51]. By carefully considering these factors, researchers can harness the full potential of DNNs in extracting meaningful insights from text data, offering a robust framework for analyzing textual information with high accuracy and efficiency. We will apply a deep neural network (DNN) algorithm after normalizing the data to the 0–1 range. The DNN algorithm will be used to predict remission to each antidepressant individually rather than grouping them into classes (e.g., SSRIs, SNRIs), facilitating personalized treatment recommendations. Missing data will be addressed using appropriate imputation techniques, chosen based on the dataset’s characteristics. Our feature selection process will encompass a wide array of variables, including genetic, epigenetic, and non-molecular biomarkers, selected through correlation analysis and domain knowledge-driven methods. Hyperparameter tuning will optimize model performance using systematic approaches like grid search. To combat overfitting, we will incorporate batch normalization and dropout layers alongside early stopping and cross-validation techniques. These strategies aim to develop a robust and generalizable DNN model for predicting treatment response in MDD patients, enhancing its applicability in real-world clinical settings.

We will split the data into training and test sets with a 90/10 ratio. DNN models will be constructed by adjusting hyperparameters (such as multiple layers, activation, etc.). To control overfitting and model convergence, we will use a batch normalization technique. Finally, we will compare DNN models to select the one that has the best confidence in predicting outcomes.

When working with a dataset of 350 individuals, dividing it into training, validation, and prediction sets is common in machine learning. The training set is used to train the model, the validation set aids in fine-tuning hyperparameters and preventing overfitting, while the prediction set evaluates the model’s performance on unseen data [52]. However, this approach has limitations due to reduced sample sizes in each subset, potentially affecting the model’s ability to generalize. With approximately 120 patients per group, concerns arise about data representativeness and the model’s reliability. Moreover, the smaller dataset size may limit the complexity of effectively trained models, as larger datasets are needed for learning meaningful patterns. Thus, while dividing the dataset is crucial, researchers must be mindful of these limitations, adapting their modeling approach accordingly to account for smaller sample sizes.

Validation datasets, separate from training data, are essential for evaluating a model’s generalizability. They assess how well the model performs on unseen data, which is crucial for accurate predictions [53]. Small datasets, like ours (350 samples), make validation even more important. Techniques like cross-validation help by dividing the data for iterative training and validation, reducing the risk of overfitting and providing a more reliable performance estimate. External validation datasets from independent studies or real-world sources further strengthen the model [54]. They test its performance in diverse settings, ensuring generalizability beyond the training data. Discussing validation approaches and including external datasets enhance research credibility. It demonstrates the model’s ability to predict patient status using new data, boosting its potential for real-world clinical application.

To assess the efficacy and reliability of AI prediction tools, quantifiable aspects and performance metrics are essential. These include accuracy, measuring correct predictions overall; precision, capturing true positives among positive predictions; recall, assessing true positives among actual positives; and F1-Score, balancing precision and recall. Additionally, the AUC-ROC evaluates the model distinction between positive and negative instances, while MSE (mean squared error) and R-squared gauge predictive accuracy in regression tasks [55]. Calibration curves check the alignment between predicted and actual probabilities. These metrics provide an objective evaluation of model performance, guiding refinement and optimization efforts for enhanced predictive accuracy and clinical utility. Systematically measuring these aspects ensures a thorough assessment of AI prediction tools’ meaningfulness, supporting reliable and generalizable outcomes Figure 2. 

### 5.2. Data Protection

The raw “omics” data will be analyzed and integrated with the medical history collected from enrollment onwards, respecting the security and privacy measures provided for in the European regulation GDPR 679/2016.

## 6. Discussion

MDD is the most common mental disorder worldwide [56]. Diagnosis of the disease is hampered by the lack of reliable biomarkers, and treatment selection requires trial and error to identify the right drug to administer to achieve remission. In addition, there are no tailored treatments for pediatric patients. The OPADE project will evaluate the interplay between genetic, epigenetic, environmental, and inflammatory factors to improve decision-making regarding assessment, accurate diagnosis, and appropriate treatment, considering possible alternative patient responses, observing outcomes, and adjusting processes accordingly. In addition to canonical medical methodology, wearable digital solutions that are useful for monitoring patient behavior, cognitive response, emotions, and social interactions are an important way to better support patients, enhance precision psychiatry, and identify physiological biomarkers and predictors of response across treatment options. The enrolled will also be encouraged to use a chatbot where they can share their mood with each other. This tool can provide data that will shed light on which social components affect individuals, more predominantly in the development of MDD. A natural language processing algorithm will then translate the conversations into machine-readable data, which will be used to derive statistical and machine-learning analyses. In addition, real-time brain activity will be monitored by a wearable EEG. The collected data will be processed by artificial intelligence and translated into different mental states. In summary, the OPADE project aims to analyze MDDs at different levels and with different approaches, ultimately leading to the constitution of a machine learning algorithm capable of improving diagnosis and treatment, preventing relapse. Machine learning algorithms offer significant assistance in diagnosing and prognosticating major depressive disorder (MDD) despite its uniform presence among subjects. These algorithms perform several crucial functions: subtyping MDD or assessing its severity using features like symptom profiles and biomarkers, enabling tailored treatment strategies. They predict treatment responses based on genetic profiles and clinical data, offering personalized plans. Additionally, they detect comorbidities such as anxiety disorders early, improving overall management. Furthermore, machine learning algorithms longitudinally monitor patients, tracking disease progression and relapse risk for timely interventions. They also stratify patients by risk, facilitating targeted prevention of adverse outcomes like suicide attempts. Overall, despite the homogeneity of MDD presence among subjects, machine learning algorithms significantly enhance diagnostic accuracy, optimize treatment selection, and improve patient outcomes through personalized and data-driven approaches.

The project is expected to lead to important discoveries that will revolutionize the clinical management of MDDs and the well-being of patients affected by the disease.

## 7. Ethics

The study was approved by the respective national regulatory authorities at all centers. All patients will provide written informed consent prior to enrollment. This project will consider ethics throughout the project and across all WPs. The consortium will adopt the EC Ethics Guidelines of the appointed ethics committees and coordinate their work by transmitting feedback to the relevant WPs. The data management plan for data, security, and privacy was implemented through interaction with hospital safety specialists and ethics committees. It will be monitored with appropriate surveys and monthly meetings with all partners.

The key to the subjects’ code is in the hands of the doctors, and the researcher sees pseudonymous (encoded) data. To prevent access to pseudonymous search data by unauthorized persons, the search data will be stored as computer files bearing a secret code not accessible to anyone except the investigators in charge of the work. The subject name and sample code are not reconciled at any stage of the study.

The data Pseudonymization is structured and applied as required by local regulatory requirements. All personal data shall be processed in such a way as not to unduly prejudice the rights and freedoms of the data subjects. Each clinical partner will obtain the necessary approvals for their work from the ethics committees.

For the processing of data in statistical models of artificial intelligence, we will follow the concepts of data governance outlined in recitals 71 of the GDPR, Article 4, and Articles 13, 14, and 22.

## Figures and Tables

**Figure 1 brainsci-14-00658-f001:**
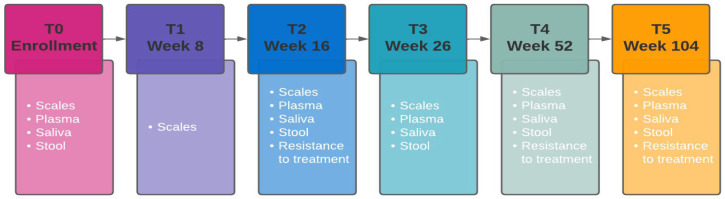
Schematic overview of the data and sample collection procedure in the OPADE (optimize and predict antidepressant efficacy for patients with major depressive disorders using multi-omics analysis and AI-predictive tool) study.

**Figure 2 brainsci-14-00658-f002:**
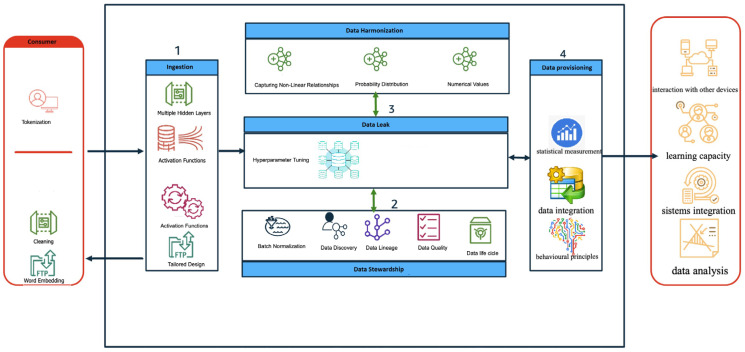
Schematic overview of artificial intelligence modeling with deep neural networks (DNNs).

**Table 1 brainsci-14-00658-t001:** Inclusion and exclusion criteria.

Inclusion criteria: Patients diagnosed with major depressive disorder (as certified by a SCID 5 for DSM-S for adults and K-SADS-PL-DSM 5 for adolescents); Currently experiencing a major depressive episode with a HAM-D score of 18 or greater, or alternatively, a MADRS score of 18 or greater; About to start a new antidepressant; Not concurrently starting a new psychotropic medication except a new antidepressant; Age 14–50 years; Able to use mobile devices (smartphone, tablet); Willingness to provide written informed consent to participate.
Exclusion criteria: Neurological disease (multiple sclerosis, severe neurocognitive disorder, epilepsy); Current psychotic disorder or mood disorder with psychotic features; Primary diagnosis of alcohol or substance use disorder (DSM-5); Patients who started concomitant psychotropic medications less than one week ago; Active, ongoing inflammatory diseases (such as rheumatoid arthritis and rheumatic polymyalgia) or severe and unstable physical illness (such as recent myocardial infarction); A history of hepatitis B or C, human immunodeficiency virus, or evidence of active tuberculosis infection or any active systemic infection within 2 weeks prior to the start of the study; Use of antibiotics or other medications that may have affected the composition of the microbiota during the 30 days prior to baseline; Pregnancy and lactation.

## Data Availability

Data are contained within the article.

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
