# Peer review of "Optimizing and Predicting Antidepressant Efficacy in Patients with Major Depressive Disorder Using Multi-Omics Analysis and the Opade AI Prediction Tools"

_brainsci, 2024, doi:10.3390/brainsci14070658_

Round 1

Reviewer 1 Report

Comments and Suggestions for Authors

Dear authors,

I found the study design titled "OPTIMIZING AND PREDICTING ANTIDEPRESSANT EFFICACY IN PATIENTS WITH MAJOR DEPRESSIVE DISORDER USING MULTI-OMICS ANALYSIS AND THE OPADE AI PREDICTION TOOLS" by Corrivetti et al to be highly comprehensive and engaging to read. However, I believe there is room for further improvement, particularly regarding the AI system aspect. Here are some specific suggestions to enhance clarity and understanding for readers:

  1. Specify Input, Output, and Network Design: it might be a good idea to provide more specific details about the AI system, including the input data utilized, the nature of the output generated, and the rationale behind the network design. Clarifying why the chosen network architecture is suitable for the task at hand would also be important to mention,

  1. Training and Validation Datasets: it could be important to Clearly delineate the training datasets used, particularly in light of the relatively small dataset size of 350 individuals. If there is no public data with all the input parameters that you will gather than you might need to divide your records into three (tranining, validation and prediction) that would mean around 120 patients per group..would you consider this a limitation?

  2.  
  3. otherwise, you can also elaborate on any validation datasets employed and consider discussing the inclusion of external validation datasets to increase the robustness of the findings and examine the ability of the model to predict patient status using unseen data

  4.  
  5. Quantifiable Aspects of Evaluation: it would be great if you can outline the quantifiable aspects that will be measured to assess the meaningfulness of the data produced by the generated network. This could include metrics such as accuracy, precision, recall, or other relevant performance indicators to evaluate the efficacy and reliability of the AI prediction tools.

  6. Also including more figures to illustrate the study design as a whole, and the AI component would greatly aid readers in grasping the overall framework of your research.

Comments on the Quality of English Language

The manuscript needs to be checked for typos.

Author Response

Dear authors,

I found the study design titled "OPTIMIZING AND PREDICTING ANTIDEPRESSANT EFFICACY IN PATIENTS WITH MAJOR DEPRESSIVE DISORDER USING MULTI-OMICS ANALYSIS AND THE OPADE AI PREDICTION TOOLS" by Corrivetti et al to be highly comprehensive and engaging to read. However, I believe there is room for further improvement, particularly regarding the AI system aspect. Here are some specific suggestions to enhance clarity and understanding for readers:

  1. Specify Input, Output, and Network Design: it might be a good idea to provide more specific details about the AI system, including the input data utilized, the nature of the output generated, and the rationale behind the network design. Clarifying why the chosen network architecture is suitable for the task at hand would also be important to mention,

Text analysis with Deep Neural Networks (DNNs) involves preprocessing textual data through tokenization, cleaning, and word embedding to convert it into numerical form. The DNN architecture, with its multiple hidden layers and activation functions, is meticulously designed to extract intricate patterns and relationships within the data. This architecture is tailored based on dataset characteristics and specific task requirements. DNNs excel in capturing complex non-linear relationships, making them suitable for text analysis tasks. They can produce probability distributions for classification tasks and numerical values for regression tasks. Additional techniques like Batch Normalization and hyperparameter tuning are employed to enhance performance and stability during training. By carefully considering these factors, researchers can harness the full potential of DNNs in extracting meaningful insights from text data, offering a robust framework for analyzing textual information with high accuracy and efficiency.

  1. Training and Validation Datasets: it could be important to Clearly delineate the training datasets used, particularly in light of the relatively small dataset size of 350 individuals. If there is no public data with all the input parameters that you will gather than you might need to divide your records into three (tranining, validation and prediction) that would mean around 120 patients per group..would you consider this a limitation?

When working with a dataset of 350 individuals, dividing it into training, validation, and prediction sets is common in machine learning. The training set is used to train the model, the validation set aids in fine-tuning hyperparameters and preventing overfitting, while the prediction set evaluates the model's performance on unseen data. However, this approach has limitations due to reduced sample sizes in each subset, potentially affecting the model's ability to generalize. With approximately 120 patients per group, concerns arise about data representativeness and the model's reliability. Moreover, the smaller dataset size may limit the complexity of effectively trained models, as larger datasets are needed for learning meaningful patterns. Thus, while dividing the dataset is crucial, researchers must be mindful of these limitations, adapting their modeling approach accordingly to account for smaller sample sizes.

  1. otherwise, you can also elaborate on any validation datasets employed and consider discussing the inclusion of external validation datasets to increase the robustness of the findings and examine the ability of the model to predict patient status using unseen data

Validation datasets, separate from training data, are essential for evaluating a model's generalizability. They assess how well the model performs on unseen data, crucial for accurate predictions.

Small datasets, like ours (350 samples), make validation even more important. Techniques like cross-validation help by dividing the data for iterative training and validation, reducing the risk of overfitting and providing a more reliable performance estimate.

External validation datasets, from independent studies or real-world sources, further strengthen the model. They test its performance in diverse settings, ensuring generalizability beyond the training data.

Discussing validation approaches and including external datasets enhance research credibility. It demonstrates the model's ability to predict patient status using new data, boosting its potential for real-world clinical application

  1. Quantifiable Aspects of Evaluation: it would be great if you can outline the quantifiable aspects that will be measured to assess the meaningfulness of the data produced by the generated network. This could include metrics such as accuracy, precision, recall, or other relevant performance indicators to evaluate the efficacy and reliability of the AI prediction tools.

To assess the efficacy and reliability of AI prediction tools, quantifiable aspects and performance metrics are essential. These include accuracy, measuring correct predictions over all; precision, capturing true positives among positive predictions; recall, assessing true positives among actual positives; and F1-Score, balancing precision and recall. Additionally, the AUC-ROC evaluates model distinction between positive and negative instances, while MSE (Mean Squared Error) and R-squared gauge predictive accuracy in regression tasks. Calibration curves check alignment between predicted and actual probabilities. These metrics provide objective evaluation of model performance, guiding refinement and optimization efforts for enhanced predictive accuracy and clinical utility. Systematically measuring these aspects ensures thorough assessment of AI prediction tools' meaningfulness, supporting reliable and generalizable outcomes.

  1. Also including more figures to illustrate the study design as a whole, and the AI component would greatly aid readers in grasping the overall framework of your research.

FIGURE 2

Master data platform – AI component.

FIGURE 3

STUDY DESIGN

Reviewer 2 Report

Comments and Suggestions for Authors

This is a very interesting manuscript that describes a proposed very complex non-profit, observational, multi-center, open-label study in individuals with Major Depressive Disorder that aims to gain fundamental insights into the role of physiological pathways involved in the onset and development of the disease. The manuscript is generally written in a careful and understandable manner.  The manuscript/protocol would benefit from consideration of several issues.

General comments:

1.     While DNA methylation signatures may have been proposed as a means of monitoring antidepressant response, where is the evidence that this approach has been successful?

2.     Why not assess resistance to treatment at baseline?  It is possible that some participants already meet criteria before enrollment in this trial.

3.     Will the psychometric rating scales be used in the patient's native language?  Please provide references that document validation of each scale in the languages to be used.

4.     The use of multiple rating scales suggests that the participants could be subject to interview fatigue. What steps will be used to prevent/mitigate this?

5.     There are no references given on the validity or reliability of using a “chatbot” to collect information about social interactions in depressed patients.

6.     A power analysis is not presented to determine the size of an adequate sample to detect changes in the primary outcome measures.

Specific Comments:

1.     Line 100, strike "without a satisfactory response", it is redundant.

2.     Have the translated forms of the CANTAB been validated?  If so, give a reference(s).

3.     Line 318, the time points for sample assessments are described as M2, M4, M6, M12, and M24. In Figure 1, these are labeled as T1, T2, ... T5.  The authors should be consistent in terminology.

4.     Line 332 states “peer support groups have been shown to 332 reduce states of anxiety, helplessness, confusion and depression”, but no reference is provided.

5.     Line 446, “DDN” should be “DNN”.

Comments on the Quality of English Language

Minor recommendations for corrections were included in my comments.

Author Response

This is a very interesting manuscript that describes a proposed very complex non-profit, observational, multi-center, open-label study in individuals with Major Depressive Disorder that aims to gain fundamental insights into the role of physiological pathways involved in the onset and development of the disease. The manuscript is generally written in a careful and understandable manner.  The manuscript/protocol would benefit from consideration of several issues.

General comments:

  1. While DNA methylation signatures may have been proposed as a means of monitoring antidepressant response, where is the evidence that this approach has been successful?

Would refer to this article:

The Relationship between DNA Methylation and Antidepressant Medications: A Systematic Review - PMC (nih.gov)

  1. Why not assess resistance to treatment at baseline?  It is possible that some participants already meet criteria before enrollment in this trial.

Several factors may hinder the assessment of resistance to treatment at baseline. Firstly, the lack of standardized guidelines or assessment tools for treatment resistance in MDD could lead to inconsistent results. Secondly, the study's primary focus on identifying predictive biomarkers and implementing personalized treatments might divert resources from assessing treatment resistance. Thirdly, practical considerations such as prolonged enrollment and increased burden on researchers and participants may not align with the study's timeline and goals. Lastly, ethical concerns regarding delayed access to potentially effective treatments for identified resistant participants may prioritize timely interventions based on current clinical needs. Thus, while assessing treatment resistance at baseline could be insightful, it may not be feasible or aligned with the OPADE study's objectives and practical considerations, which emphasize personalized treatment approaches for MDD.

Thanks to your suggestion. We will evaluate it prospectively with a specific tool. We will try to recover the information relating to resistance in previous depressive episodes, for possible secondary analysis.

  1. Will the psychometric rating scales be used in the patient's native language?  Please provide references that document validation of each scale in the languages to be used.

Trajković, G., Starčević, V., Latas, M., Leštarević, M., Ille, T., Bukumirić, Z., & Marinković, J. (2011). Reliability of the Hamilton Rating Scale for Depression: a meta-analysis over a period of 49 years. Psychiatry research, 189(1), 1–9. https://doi.org/10.1016/j.psychres.2010.12.007

Nuevo, R., Dunn, G., Dowrick, C., Vázquez-Barquero, J. L., Casey, P., Dalgard, O. S., Lehtinen, V., & Ayuso-Mateos, J. L. (2009). Cross-cultural equivalence of the Beck Depression Inventory: a five-country analysis from the ODIN study. Journal of affective disorders, 114(1-3), 156–162. https://doi.org/10.1016/j.jad.2008.06.021

Williams, J. B., & Kobak, K. A. (2008). Development and reliability of a structured interview guide for the Montgomery Asberg Depression Rating Scale (SIGMA). The British journal of psychiatry : the journal of mental science, 192(1), 52–58. https://doi.org/10.1192/bjp.bp.106.032532

Ioannou, M., Dellepiane, M., Benvenuti, A., Feloukatzis, K., Skondra, N., Dell'Osso, L., & Steingrímsson, S. (2016). Swedish Version of Mood Spectrum Self-Report Questionnaire: Psychometric Properties of Lifetime and Last-week Version. Clinical practice and epidemiology in mental health : CP & EMH, 12, 14–23. https://doi.org/10.2174/1745017901612010014

Boyd, J. E., Adler, E. P., Otilingam, P. G., & Peters, T. (2014). Internalized Stigma of Mental Illness (ISMI) scale: a multinational review. Comprehensive psychiatry, 55(1), 221–231. https://doi.org/10.1016/j.comppsych.2013.06.005

Aas I. H. (2010). Global Assessment of Functioning (GAF): properties and frontier of current knowledge. Annals of general psychiatry, 9, 20. https://doi.org/10.1186/1744-859X-9-20

Hanssen-Bauer, K., Gowers, S., Aalen, O. O., Bilenberg, N., Brann, P., Garralda, E., Merry, S., & Heyerdahl, S. (2007). Cross-national reliability of clinician-rated outcome measures in child and adolescent mental health services. Administration and policy in mental health, 34(6), 513–518. https://doi.org/10.1007/s10488-007-0135-x

Wagner, A. K., Gandek, B., Aaronson, N. K., Acquadro, C., Alonso, J., Apolone, G., Bullinger, M., Bjorner, J., Fukuhara, S., Kaasa, S., Leplège, A., Sullivan, M., Wood-Dauphinee, S., & Ware, J. E., Jr (1998). Cross-cultural comparisons of the content of SF-36 translations across 10 countries: results from the IQOLA Project. International Quality of Life Assessment. Journal of clinical epidemiology, 51(11), 925–932. https://doi.org/10.1016/s0895-4356(98)00083-3

Varni, J.W., Seid, M. & Kurtin, P.S. (2001). PedsQL 4.0: Reliability and Validity of the Pediatric Quality of Life Inventory Version 4.0 Generic Core Scales in Healthy and Patient Populations. Medical Care, 39(8), 800-812.

Craig, C. L., et al. (2003). "International physical activity questionnaire: 12-country reliability and validity." Med Sci Sports Exerc 35: 1381-95.

Ravens-Sieberer, U., Herdman, M., Devine, J., Otto, C., Bullinger, M., Rose, M., & Klasen, F. (2014). The European KIDSCREEN approach to measure quality of life and well-being in children: development, current application, and future advances. Quality of life research : an international journal of quality of life aspects of treatment, care and rehabilitation, 23(3), 791–803. https://doi.org/10.1007/s11136-013-0428-3

McIntyre, R. S., Alsuwaidan, M., Baune, B. T., Berk, M., Demyttenaere, K., Goldberg, J. F., Gorwood, P., Ho, R., Kasper, S., Kennedy, S. H., Ly-Uson, J., Mansur, R. B., McAllister-Williams, R. H., Murrough, J. W., Nemeroff, C. B., Nierenberg, A. A., Rosenblat, J. D., Sanacora, G., Schatzberg, A. F., Shelton, R., … Maj, M. (2023). Treatment-resistant depression: definition, prevalence, detection, management, and investigational interventions. World psychiatry: official journal of the World Psychiatric Association (WPA), 22(3), 394–412. https://doi.org/10.1002/wps.21120

  1. The use of multiple rating scales suggests that the participants could be subject to interview fatigue. What steps will be used to prevent/mitigate this?

In optimizing the administration of rating scales in our study, several strategies will enhance participant experience and data quality. First, implementing a rotation of scales across participants will help to distribute cognitive load and prevents monotony-induced fatigue. Additionally, incorporating breaks between scale administrations allows participants to recharge mentally, mitigating cognitive overload. Moreover, soliciting feedback from participants about their experience with the scales, including fatigue levels and suggestions for improvement, fosters participant engagement and improves protocol effectiveness. Creating a supportive environment with comfortable seating, adequate lighting, and privacy minimizes distractions and promotes focus during scale administration. Lastly, providing comprehensive training and ongoing support to research staff ensures consistent and accurate scale administration, reducing the need for repeated assessments and enhancing data reliability. By implementing these strategies, we aim to optimize the rating scale administration process, ultimately enhancing the validity and reliability of our study outcomes.

  1. There are no references given on the validity or reliability of using a “chatbot” to collect information about social interactions in depressed patients.

Schick, A., Feine, J., Morana, S., Maedche, A., & Reininghaus, U. (2022). Validity of Chatbot Use for Mental Health Assessment: Experimental Study. JMIR mHealth and uHealth, 10(10), e28082. https://doi.org/10.2196/28082

Abd-Alrazaq, A. A., Rababeh, A., Alajlani, M., Bewick, B. M., & Househ, M. (2020). Effectiveness and Safety of Using Chatbots to Improve Mental Health: Systematic Review and Meta-Analysis. Journal of medical Internet research, 22(7), e16021. https://doi.org/10.2196/16021

de Gennaro, M., Krumhuber, E. G., & Lucas, G. (2020). Effectiveness of an Empathic Chatbot in Combating Adverse Effects of Social Exclusion on Mood. Frontiers in psychology, 10, 3061. https://doi.org/10.3389/fpsyg.2019.03061

  1. A power analysis is not presented to determine the size of an adequate sample to detect changes in the primary outcome measures.

Planned MVA study is a feasibility study; thus, no power calculation is needed (https://doi.org/10.1186/1471-2288-13-104)

Specific Comments:

  1. Line 100, strike "without a satisfactory response", it is redundant.

DONE

  1. Have the translated forms of the CANTAB been validated?  If so, give a reference(s).

Robbins TW, Sahakian BJ. Computer methods of assessment of cognitive function. In Principles and Practice of Geriatric Psychiatry, Copeland JRM, Abou-Saleh MT, Blazers DG (eds), John Wiley & Sons Ltd., Chichester, 1994, pp 205-209.

Torgersen, J., Flaatten, H.K., Engelsen, B.A., & Gramstad, A. (2012). Clinical Validation of Cambridge Neuropsychological Test Automated Battery in a Norwegian Epilepsy Population. Journal of Behavioral and Brain Science, 2, 108-116.

Siew, S. K. H., Han, M. F. Y., Mahendran, R., & Yu, J. (2022). Regression-Based Norms and Validation of the Cambridge Neuropsychological Test Automated Battery among Community-Living Older Adults in Singapore. Archives of clinical neuropsychology : the official journal of the National Academy of Neuropsychologists, 37(2), 457–472.

  1. Line 318, the time points for sample assessments are described as M2, M4, M6, M12, and M24. In Figure 1, these are labeled as T1, T2, ... T5.  The authors should be consistent in terminology.

DONE

  1. Line 332 states “peer support groups have been shown to 332 reduce states of anxiety, helplessness, confusion and depression”, but no reference is provided.

Richard, J., Rebinsky, R., Suresh, R., Kubic, S., Carter, A., Cunningham, J. E. A., Ker, A., Williams, K., & Sorin, M. (2022). Scoping review to evaluate the effects of peer support on the mental health of young adults. BMJ open, 12(8), e061336. https://doi.org/10.1136/bmjopen-2022-061336

  1. Line 446, “DDN” should be “DNN”.

DONE

Reviewer 3 Report

Comments and Suggestions for Authors

The authors present a protocol for an innovative multisite longitudinal observational study in patients with major depression. The study will enroll 350 patients with MDD across 5 countries and collect biological, cognitive, and clinical data that will be used to train deep neural networks for identification of predictive biomarkers of treatment response. There is a pressing need to identify such biomarkers, but the following concerns need to be address: 

Overall:

-The manuscript requires additional citations in multiple places, some of which have been pointed out below.

Abstract:

-Line 45: The WHO report in 2017 had MDD as the leading cause of disability. Are you referencing a more updated report?

-Line 53 stating that the study "aims to gain fundamental insights into the role of physiological pathways involved in the onset and development of the disease" doesn't align with the objective stated in the Introduction or the fact that the patients will already have MDD at enrollment. 

Introduction:

-In line 64-65, it is stated that MDD is the most common mental health disorder, but anxiety disorders are the most common according to the 2019 WHO report: https://www.who.int/news-room/fact-sheets/detail/mental-disorders

-For lines 65-67, I'd recommend listing criteria according to the DSM-5-TR or ICD-11

-Sentence in lines 101-103 should have a citation

-Line 108: "amnestic" does not appear to be the correct word choice

-Line 118: "anamesis" does not appear to be the correct word choice

Study Design:

Table 1: Consider clarifying the criteria "Not concurrently starting a new psychotropic medication" by changing to "Not concurrently starting a new psychotropic medication except a new antidepressant"

Lines 216-217: This information should appear in a separate section and the abbreviations should be spelled out

Lines 333-335: This sentence needs additional explanation: "Through mathematical models, patients' stories will be transformed into data accessible to the entire research group."

-Line 338: "Chains of environmental factors..." needs to be reworded

-Line 356: "host" should be changed to "patient" or "individual".

-Lines 360-361: Needs citation and effect sizes. Would be helpful to have some examples of how these effect sizes compare to other biological measures. 

-Lines 362-363: Needs citation

-Line 381: Aims 1-7 should be defined previously

Section 5.1: When groups are discussed, is this referring to age groups, groups taking the same antidepressants, or both types of groups? Will each antidepressant be assessed individually or will they be grouped (e.g., SSRIs, SNRIs, etc.)?

-Section 5.2 should be expanded to include more details. Will the algorithm be used to predict remission to each antidepressant individually or will they be grouped (e.g., SSRIs, SNRIs, etc.)? How will missing data be handled? Which features will be inputted into the model and what method will be used for feature selection? What methods will be used for hyperparameter tuning? What will be done to address overfitting?

Discussion:

Line 443: How will the ML algorithm be used to help with diagnosis when all subjects have MDD?

Ethics

Line 446: Additional details should be provided about data security and patient privacy. 

Comments on the Quality of English Language

The manuscript should be reviewed for appropriate word choice. 

Author Response

The authors present a protocol for an innovative multisite longitudinal observational study in patients with major depression. The study will enroll 350 patients with MDD across 5 countries and collect biological, cognitive, and clinical data that will be used to train deep neural networks for identification of predictive biomarkers of treatment response. There is a pressing need to identify such biomarkers, but the following concerns need to be address: 

Overall:

-The manuscript requires additional citations in multiple places, some of which have been pointed out below.

DONE

Abstract:

-Line 45: The WHO report in 2017 had MDD as the leading cause of disability. Are you referencing a more updated report?

Institute of Health Metrics and Evaluation. Global Health Data Exchange (GHDx). https://vizhub.healthdata.org/gbd-results/ (Accessed 4 March 2023).

-Line 53 stating that the study "aims to gain fundamental insights into the role of physiological pathways involved in the onset and development of the disease" doesn't align with the objective stated in the Introduction or the fact that the patients will already have MDD at enrollment. 

Here, we describe a multicentre, observational, open-label, non-profit study conducted in subjects with MDD, with the aim of gaining fundamental knowledge about the role of physiological pathways involved in the development and outcome of the disease. 350 patients aged between 14 and 50 years will be recruited in 6 Countries (Italy, Colombia, Spain, The Netherlands, Turkey) for 24 months

Introduction:

-In line 64-65, it is stated that MDD is the most common mental health disorder, but anxiety disorders are the most common according to the 2019 WHO report: https://www.who.int/news-room/fact-sheets/detail/mental-disorders

It's true. It should be modified as follows: it is the second most frequent mental disorder worldwide, after anxiety disorder

-For lines 65-67, I'd recommend listing criteria according to the DSM-5-TR or ICD-11

It is necessary to anticipate what is written in lines 87-90 to 65-67

-Sentence in lines 101-103 should have a citation

Voineskos, D., Daskalakis, Z. J., & Blumberger, D. M. (2020). Management of Treatment-Resistant Depression: Challenges and Strategies. Neuropsychiatric disease and treatment, 16, 221–234. https://doi.org/10.2147/NDT.S198774

-Line 108: "amnestic" does not appear to be the correct word choice 

DONE

-Line 118: "anamesis" does not appear to be the correct word choice

DONE

Study Design:

Table 1: Consider clarifying the criteria "Not concurrently starting a new psychotropic medication" by changing to "Not concurrently starting a new psychotropic medication except a new antidepressant"

DONE

Lines 216-217: This information should appear in a separate section and the abbreviations should be spelled out   

DONE

Lines 333-335: This sentence needs additional explanation: "Through mathematical models, patients' stories will be transformed into data accessible to the entire research group."

Traditionally, patient narratives have been invaluable qualitative data in healthcare research, yet analyzing and sharing this information effectively across teams has posed challenges. However, a groundbreaking approach has emerged: harnessing mathematical models to bridge this gap.

By converting patients' stories into quantitative data, a new level of comprehension is unlocked. These models capture qualitative elements such as emotions and challenges as measurable data points, enabling collaborative analysis among diverse research team members.

Picture psychologists, data scientists, and clinicians working together to analyze quantitative data derived from patient narratives. This not only enhances data accessibility but also fosters communication and collaboration within the research group

-Line 338: "Chains of environmental factors..." needs to be reworded

DONE

-Line 356: "host" should be changed to "patient" or "individual".

DONE

-Lines 360-361: Needs citation and effect sizes. Would be helpful to have some examples of how these effect sizes compare to other biological measures. 

Examples of the effects of the sample size and of how these compare to other biological measures have been recently and extensively reviewed [Yuan M, Yang B, Rothschild G, Mann JJ, Sanford LD, Tang X, Huang C, Wang C, Zhang W. Epigenetic regulation in major depres-sion and other stress-related disorders: molecular mechanisms, clinical relevance and therapeutic potential. Signal Transduct Target Ther. 2023 Aug 30;8(1):309. doi: 10.1038/s41392-023-01519-z. PMID: 37644009; PMCID: PMC10465587.]

-Lines 362-363: Needs citation

Webb, L. M., Phillips, K. E., Ho, M. C., Veldic, M., & Blacker, C. J. (2020). The Relationship between DNA Methylation and Antidepressant Medications: A Systematic Review. International journal of molecular sciences, 21(3), 826. https://doi.org/10.3390/ijms21030826

Powell, T. R., Smith, R. G., Hackinger, S., Schalkwyk, L. C., Uher, R., McGuffin, P., Mill, J., & Tansey, K. E. (2013). DNA methylation in interleukin-11 predicts clinical response to antidepressants in GENDEP. Translational psychiatry, 3(9), e300. https://doi.org/10.1038/tp.2013.73

-Line 381: Aims 1-7 should be defined previously

DONE

Section 5.1: When groups are discussed, is this referring to age groups, groups taking the same antidepressants, or both types of groups? Will each antidepressant be assessed individually or will they be grouped (e.g., SSRIs, SNRIs, etc.)?

We referred to age groups and the antidepressant will be assessed individually

-Section 5.2 should be expanded to include more details. Will the algorithm be used to predict remission to each antidepressant individually or will they be grouped (e.g., SSRIs, SNRIs, etc.)? How will missing data be handled? Which features will be inputted into the model and what method will be used for feature selection? What methods will be used for hyperparameter tuning? What will be done to address overfitting?

The DNN algorithm will be used to predict remission to each antidepressant individually, rather than grouping them into classes (e.g., SSRIs, SNRIs), facilitating personalized treatment recommendations. Missing data will be addressed using appropriate imputation techniques, chosen based on the dataset's characteristics. Our feature selection process will encompass a wide array of variables, including genetic, epigenetic, and non-molecular biomarkers, selected through correlation analysis and domain knowledge-driven methods. Hyperparameter tuning will optimize model performance using systematic approaches like grid search. To combat overfitting, we'll incorporate batch normalization and dropout layers, alongside early stopping and cross-validation techniques. These strategies aim to develop a robust and generalizable DNN model for predicting treatment response in MDD patients, enhancing its applicability in real-world clinical settings.

Discussion:

Line 443: How will the ML algorithm be used to help with diagnosis when all subjects have MDD?

Machine learning algorithms offer significant assistance in diagnosing and prognosticating Major Depressive Disorder (MDD), despite its uniform presence among subjects. These algorithms perform several crucial functions: subtyping MDD or assessing its severity using features like symptom profiles and biomarkers, enabling tailored treatment strategies. They predict treatment responses based on genetic profiles and clinical data, offering personalized plans. Additionally, they detect comorbidities such as anxiety disorders early, improving overall management. Furthermore, machine learning algorithms longitudinally monitor patients, tracking disease progression and relapse risk for timely interventions. They also stratify patients by risk, facilitating targeted prevention of adverse outcomes like suicide attempts. Overall, despite the homogeneity of MDD presence among subjects, machine learning algorithms significantly enhance diagnostic accuracy, optimize treatment selection, and improve patient outcomes through personalized and data-driven approaches.

Ethics

Line 446: Additional details should be provided about data security and patient privacy.

This project will consider ethics throughout the project and across all WPs. The consortium will adopt the EC Ethics Guidelines and appoint ethics committees and coordinate their work by transmitting feedback to the relevant WPs. The data management plan for data, security and privacy was implemented through interaction with hospital safety specialists and ethics committees. It will be monitored with appropriate surveys and monthly meetings with all partners.

The key to the subjects' code is in the hands of the doctors, and the researcher sees pseudonymous (encoded) data. To prevent access to pseudonymous search data by unauthorized persons, the search data will be stored as computer files bearing a secret code, not accessible to anyone except the investigators in charge of the work. Subject name and sample code are not reconciled at any stage of the study.

The data Pseudonymization is structured and applied as required by local regulatory requirements. All personal data shall be processed in such a way as not to unduly prejudice the rights and freedoms of the data subjects. Each clinical partner will obtain the necessary approvals for their work from the ethics committees.

For the processing of data in statistical models of Artificial Intelligence, we will follow the concepts of data governance outlined in recitals 71 of the GDPR, Article 4 and Articles 13 and 14 and 22.

Reviewer 4 Report

Comments and Suggestions for Authors

This study is expected to be promising, but the overall description of the manuscript seems disorganized and inconsistent.

In the introduction, it would be beneficial to introduce the application of novel research tools and clearly articulate the research objectives and its distinctiveness from previous studies.

The statistics and number related to MDD in the introduction and abstract lack consistency and clarity. Additionally, please verify the gender ratio statement regarding depression. "Depression is about 50% more common in women than in men", "female to male ratio of approximately 2:1"

Instead, discussing variations by race and country, especially since the study is expected to be conducted in multiple countries, might be helpful.

there seems to be inconsistency in the exclusion criteria, as "Intellectual disability" is also a type of "Neurological disease." It would be helpful to maintain consistency in the organization of the exclusion criteria.

Author Response

This study is expected to be promising, but the overall description of the manuscript seems disorganized and inconsistent.

In the introduction, it would be beneficial to introduce the application of novel research tools and clearly articulate the research objectives and its distinctiveness from previous studies.

DONE

The statistics and number related to MDD in the introduction and abstract lack consistency and clarity. Additionally, please verify the gender ratio statement regarding depression. "Depression is about 50% more common in women than in men", "female to male ratio of approximately 2:1"

DONE

Instead, discussing variations by race and country, especially since the study is expected to be conducted in multiple countries, might be helpful.

 DONE

there seems to be inconsistency in the exclusion criteria, as "Intellectual disability" is also a type of "Neurological disease." It would be helpful to maintain consistency in the organization of the exclusion criteria.

DONE

Round 2

Reviewer 1 Report

Comments and Suggestions for Authors

The paragraphs covering the AI analysis, still need to be referenced. 

about the validtaion step, you can write, we will try to validate our apporach using appropriate external datasets ( if you can, add there names)

Comments on the Quality of English Language

only minor corrections are needed.

Author Response

The paragraphs covering the AI analysis, still need to be referenced. About the validtaion step, you can write, we will try to validate our apporach using appropriate external datasets ( if you can, add there names)

Thank you for pointing this out. We agree with this comment. Therefore, we have modified. 
We have also added references. 

Reviewer 3 Report

Comments and Suggestions for Authors

The authors have adequately addressed all concerns except the following: the objective stated in the abstract does not align with the objective stated in the main manuscript.

Abstract: Here, we describe a non-profit, observational, multicenter, open-label study in individuals with MDD that aims to gain fundamental insights into the role of physiological pathways involved in the onset and development of the disease.

Manuscript: OPADE is a non-profit, observational, multicenter, open-label study aimed at identifying predictive biomarkers for stratification and implementation of personalized drug treatments in patients with MDD, to guide healthcare provider decision making by developing an AI/ML predictive tool that will be capable of combining genetics, epigenetics, microbiome, immune response data, and also the non-molecular biomarkers such as medical history, electroencephalography (EEG), from subjects with MDD. 

Author Response

The authors have adequately addressed all concerns except the following: the objective stated in the abstract does not align with the objective stated in the main manuscript.

Abstract: Here, we describe a non-profit, observational, multicenter, open-label study in individuals with MDD that aims to gain fundamental insights into the role of physiological pathways involved in the onset and development of the disease.

Manuscript: OPADE is a non-profit, observational, multicenter, open-label study aimed at identifying predictive biomarkers for stratification and implementation of personalized drug treatments in patients with MDD, to guide healthcare provider decision making by developing an AI/ML predictive tool that will be capable of combining genetics, epigenetics, microbiome, immune response data, and also the non-molecular biomarkers such as medical history, electroencephalography (EEG), from subjects with MDD. 

Thank you for pointing this out. We have modified the text according to your suggestions.

Reviewer 4 Report

Comments and Suggestions for Authors

Merely stating 'done' in the response letter and the absence of clear indications of changes in the manuscript file make it challenging to assess.

Author Response

Dear Editor,

Thank you for the opportunity to submit our manuscript to Brain Sciences. We appreciate the time and effort you have dedicated to providing insightful comments on our manuscript. We have made our best effort to incorporate the suggested changes and restructure the entire paper. We apologize for the mistake in uploading the wrong file in response to your first revision. Recognizing that our second reply may have been insufficient, we kindly ask you to consider the following point-by-point response to your comments and concerns.

This study is expected to be promising, but the overall description of the manuscript seems disorganized and inconsistent.

Thank you for your valuable feedback, we appreciate your positive remarks about the potential of our study. We understand your concerns regarding the organization and consistency of the manuscript. Based on your feedback, we have made several revisions to improve the clarity and structure of the manuscript. Here are the specific changes we have implemented:

  1. Enhanced Structure: We have restructured the manuscript to follow a clear and logical format, ensuring each section flows seamlessly into the next.
  2. Terminology: We have reviewed the manuscript to ensure that terminology is used consistently. Key terms are now defined clearly upon first use and consistently applied throughout the text.
  3. Formatting: We have applied a uniform format to the entire document, including consistent font type, size, and spacing also suggested by the editor.
  4. Clear Transitions:

We've meticulously incorporated clear transitions between sections and ideas throughout the manuscript. This deliberate effort serves two key purposes:

  • Improved Flow and Navigation: Transitions act as bridges, guiding the reader on a smooth journey through the research journey. By signaling connections and logical progressions, they ensure a cohesive and effortless reading experience. 
  • Enhanced Comprehension of Research: Effective transitions don't merely connect sections; but allows the reader to not only grasp each piece of information but also understand how they build upon each other to form the bigger picture.

By strategically incorporating transitions, we've ensured the reader can not only navigate the manuscript with ease but also fully comprehend the captivating story of your research and its findings.

In the introduction, it would be beneficial to introduce the application of novel research tools and clearly articulate the research objectives and its distinctiveness from previous studies.

We appreciate your suggestions on enhancing the introduction by better introducing the devices used in the study . In response to your comments, we have made the following revisions:

  1. Introduction of Novel Research Tools: Our introduction delves deeper into the innovative research tools and methodologies utilized in this study. This comprehensive explanation sheds light on the groundbreaking nature of our approach to patient stratification within major depressive disorders.
  2. Clear Articulation of Research Objectives: We have refined the introduction to clearly state the research objectives. This revision ensures that the goals of our study are explicitly outlined and easily understood by the readers.
  3. Distinctiveness from Previous Studies: This paragraph in the discussion section emphasizes the unique contributions and distinctiveness of our work.

The statistics and number related to MDD in the introduction and abstract lack consistency and clarity. Additionally, please verify the gender ratio statement regarding depression. "Depression is about 50% more common in women than in men", "female to male ratio of approximately 2:1"

Thank you for your constructive feedback regarding the statistics and numbers related to Major Depressive Disorder (MDD) reported in the manuscript. We understand the importance of consistency and clarity in presenting statistical data. Additionally, we appreciate your attention to the accuracy of the gender ratio statement concerning depression.

In response to your comments, we have made the following revisions:

  1. Consistency and Clarity of Statistics: We have reviewed and standardized all statistical data related to MDD in both the introduction and abstract. This ensures that the numbers are consistent and presented clearly throughout the manuscript.
  2. Verification of Gender Ratio Statement: We have verified the gender ratio of depression and ensured that the information is accurate and consistent across the manuscript. We now consistently state that depression is approximately twice as common in women as in men, which aligns with the commonly cited ratio of 2:1 (Bromet et al., 2011).

Instead, discussing variations by race and country, especially since the study is expected to be conducted in multiple countries, might be helpful.

Thank you for your insightful feedback and suggestion. We agree that discussing variations in Major Depressive Disorder (MDD) by race and country is crucial, particularly given the multinational scope of our study including 5 recruiting centers located in different countries.  We have revised the introduction to include a discussion of these variations, providing a broader context for our research.

There seems to be inconsistency in the exclusion criteria, as "Intellectual disability" is also a type of "Neurological disease." It would be helpful to maintain consistency in the organization of the exclusion criteria.

Thank you for your feedback regarding the exclusion criteria. We understand the importance of maintaining consistency in the organization of these criteria. To address this, we have revised the exclusion criteria to ensure clarity and consistency. Specifically, we have reorganized the criteria to clearly differentiate between various conditions, including intellectual disabilities and other neurological diseases.

Incorporating your valuable feedback, these changes aim to improve the manuscript's organization and consistency, ultimately strengthening the presentation of our study. We welcome your further comments to ensure the manuscript reaches its full potential.
